# Evidence-Based Recommendations for an Optimal Prenatal Supplement for Women in the U.S., Part Two: Minerals

**DOI:** 10.3390/nu13061849

**Published:** 2021-05-28

**Authors:** James B. Adams, Jacob C. Sorenson, Elena L. Pollard, Jasmine K. Kirby, Tapan Audhya

**Affiliations:** 1School of Engineering of Matter, Transport and Energy, Arizona State University, Tempe, AZ 85287, USA; jcsorenson20@gmail.com (J.C.S.); epollard1025@gmail.com (E.L.P.); jkkirby1@asu.edu (J.K.K.); 2Neurological Health Foundation, Dallas, TX 75230, USA; audyatk@optonline.net; 3College of Medicine, University of Arizona, Tucson, AZ 85724, USA; 4Health Diagnostics and Research Institute, South Amboy, NJ 08879, USA

**Keywords:** pregnancy, supplements, minerals, calcium, iron, magnesium, chromium, selenium

## Abstract

The levels of many essential minerals decrease during pregnancy if un-supplemented, including calcium, iron, magnesium, selenium, zinc, and possibly chromium and iodine. Sub-optimal intake of minerals from preconception through pregnancy increases the risk of many pregnancy complications and infant health problems. In the U.S., dietary intake of minerals is often below the Recommended Dietary Allowance (RDA), especially for iodine and magnesium, and 28% of women develop iron deficiency anemia during their third trimester. The goal of this paper is to propose evidence-based recommendations for the optimal level of prenatal supplementation for each mineral for most women in the United States. Overall, the evidence suggests that optimal mineral supplementation can significantly reduce a wide range of pregnancy complications (including anemia, gestational hypertension, gestational diabetes, hyperthyroidism, miscarriage, and pre-eclampsia) and infant health problems (including anemia, asthma/wheeze, autism, cerebral palsy, hypothyroidism, intellectual disability, low birth weight, neural tube defects, preterm birth, rickets, and wheeze). An evaluation of 180 commercial prenatal supplements found that they varied widely in mineral content, often contained only a subset of essential minerals, and the levels were often below our recommendations. Therefore, there is a need to establish recommendations on the optimal level of mineral supplementation during pregnancy.

## 1. Introduction

Vitamins and minerals are, by definition, essential substances that are necessary for good health, and a deficiency of any one vitamin or mineral can be serious. Although a very healthy diet rich in a wide variety of vegetables, fruits, protein, and fats can provide sufficient amounts of most vitamins and minerals, many people do not consume an adequate diet. During pregnancy, there is an increased need for vitamins and minerals to promote a healthy pregnancy and a healthy baby. Without supplementation, the levels of many minerals decrease significantly during pregnancy, including calcium, iron, magnesium, selenium, zinc, and possibly chromium and iodine.

Prenatal supplements are intended to supplement typical diets to ensure that adequate amounts of vitamins and minerals are consumed. The U.S. Food and Drug Administration (FDA) has established Recommended Dietary Allowances for total vitamin/mineral intake from food and supplements, but they have not established recommendations for prenatal supplements. Therefore, there is wide variation in the content and quality of prenatal supplements. In order to lower costs and minimize the number of pills, approximately 44% of prenatal supplements (83 out of the 188 we evaluated) consist of only a single pill with limited amounts of vitamins and often few or no minerals. This results in insufficient vitamin/mineral supplementation for many women and, hence, does not adequately protect them, or their children, from pregnancy complications and infant health problems.

The purpose of this paper is to propose a set of evidence-based recommendations of the optimal levels of each mineral for prenatal supplements. Our recommendations are based primarily on four sources:

(1) The FDA’s Recommended Daily Allowances for pregnant women, which are estimated to meet the needs of 97.5% of healthy pregnant women;

(2) The FDA’s Tolerable Upper Limit, which is the maximum amount of minerals that can be safely consumed without any risk of health problems;

(3) The National Health and Nutrition Examination Survey (NHANES), which evaluates the average intake of vitamins and minerals by women ages 20–39 years in the U.S.;

(4) Research studies on mineral deficiencies or mineral supplementation during pregnancy and their effects on pregnancy, birth, and child health problems.

In summary, the RDA (Recommended Dietary Allowance) establishes minimum recommended levels of vitamin/mineral intake from all sources, and the NHANES establishes the average intake from foods. The difference is what needs to be consumed in a supplement, on average. However, because people vary significantly in the qualities of their diets and because most minerals have a high Tolerable Upper Limit, we generally recommend more than the difference between the RDA and the average NHANES, so that most women achieve the RDA. In some cases, our recommendations exceed the RDA based on additional research studies and clinical trials, which demonstrate an increased need for those nutrients during pregnancy and benefits from higher levels of those minerals.

This paper also provides an evaluation of over 180 prenatal supplements and compares their levels against the evidence-based recommendations proposed here.

## 2. Methods

In this paper, we focus on 10 essential minerals, and each mineral is reviewed in a separate section. Each section includes a background about that mineral, a summary of research, daily intake (as estimated from the National Health and Nutrition Examination Survey), Recommended Dietary Allowance, a recommendation based on our interpretation of all this data, and statistics on prenatal supplements currently on the market.

Because the research literature is vast, a systematic review of all studies would require a separate paper on each mineral; instead, we focused on the most relevant articles that we found from keyword searches of PubMed and forward and backward citation searches of the most relevant articles. The primary focus of this review was on articles that provided insight into optimal dosage such as treatment studies on the effects of different doses on outcomes and biomarkers. Greater consideration was given to larger studies with a more rigorous design such as randomized, double-blind, placebo-controlled studies. When available, we included meta-analyses and systematic reviews of the literature; however, the limitation of those studies was that they generally asked whether or not a symptom was related to a mineral deficiency or improved due to mineral supplementation but generally did not attempt to estimate the optimal level of supplementation. The types of articles reviewed generally fell into three categories: (1) the associations of low levels of minerals with health problems, (2) studies of changes in mineral levels during pregnancy if un-supplemented or supplemented, and (3) clinical trials on the effect of mineral supplementation on health problems. See Appendix A for more information about the studies discussed in this review.

The NHANES data listed in this paper is for dietary intake only (not supplements) because we assume that most women will stop other vitamin/mineral supplements when they start a prenatal supplement. We report the data for women ages 20–39 years because that is the most common time for pregnancy, and averages for other ages are generally similar. We use the 2017–2018 NHANES data for the minerals reported then, and otherwise report the 2009–2010 data.

The ultimate goal of this review is to propose evidence-based recommendations for the optimal level of each mineral for a prenatal supplement based on currently available information, with the understanding that further research is needed for most minerals to fine-tune our recommendations. A key point is trying to balance the benefit of additional supplementation for those women with the lowest levels of minerals vs. the risk of adverse effects for women with the highest levels of minerals. No single formulation is ideal for every person. However, because personalized testing to determine individualized prenatal supplementation is rare, we believe it is important to develop evidence-based recommendations for the general population while encouraging physicians and nutritionists to personalize recommendations to the extent possible.

A comprehensive list of 188 prenatal supplements currently on the market was created primarily using two databases created by the National Institutes of Health (NIH): the Dietary Supplement Label Database (DSLD) and DailyMed. Although both databases include an extensive list of prenatal supplements, some products listed are outdated and can no longer be purchased or have changed ingredients. Therefore, the list was updated using information on manufacturer websites (when available) or from labels on retail websites such as Amazon. The contents of these prenatal supplements were then analyzed and compared against the evidence-based recommendations proposed here.

## 3. Results

### 3.1. Calcium

#### 3.1.1. Research

Calcium is essential for bone and tooth growth, so a lack of calcium in infants causes growth delays and bone deformations, otherwise known as rickets [1,2]. Calcium is also important for control of blood pressure, nerve impulses, muscle contraction, and secretion of some hormones.

Low calcium is especially associated with preeclampsia during pregnancy. Women under the age of 20, as well as women who live in the southern part of the United States, are reported to have a greater risk for preeclampsia [3]. Levels of total serum calcium and bone density decline throughout pregnancy, indicating a need for calcium supplementation for most pregnant women [4,5,6].

A meta-analysis of 14 articles compared patients who had pregnancy-induced hypertension (PIH) with patients with healthy pregnancies, and found that patients with PIH had slightly lower calcium levels than healthy gravidas [7].

A Cochrane review [8] of calcium supplementation found the following for high-dose and low-dose supplementation: For high-dose calcium supplementation (≥1 g/day), 13 high-quality studies (involving 15,730 women) found that supplementation significantly reduced the risk of high blood pressure (RR (Relative Risk) = 0.65) and preeclampsia (RR = 0.45). The effect was most significant for women with low-calcium diets and women at high risk for preeclampsia. Calcium supplementation also somewhat reduced the risk of maternal death or serious morbidity (RR = 0.80) and preterm birth (RR = 0.76). One possible rare negative effect of calcium supplementation involved two trials involving 12,901 women that found a small increased risk of HELLP (hemolysis, elevated liver enzymes, and low platelets) occurring in 16 cases among the supplemented women vs. 6 in the un-supplemented group; these two studies involved dosages of 1500–2000 mg/day of elemental calcium. Regarding childhood outcomes, one study showed a reduction in childhood elevated blood pressure, and one study found a reduced rate of dental caries at age 12.

For low-dose calcium supplementation (<1 g/day), 10 trials (involving 2234 women) evaluated supplementation with low doses of calcium alone (4 trials) or in association with vitamin D (3 trials), linoleic acid (2 trials), or antioxidants (1 trial). Supplementation with low doses of calcium substantially reduced the risk of preeclampsia (RR = 0.38). There was also a reduction in hypertension, low birth weight, and neonatal intensive care unit admission.

Similarly, another meta-analysis [9] of 27 trials with 28,492 pregnant women found that calcium supplementation was effective in decreasing the risk of preeclampsia (RR 0.51, 95% CI (Confidence Interval): 0.40 to 0.64) and gestational hypertension (RR 0.70, 95% CI: 0.60 to 0.82).

Two small studies found that 600 mg of calcium plus 450 mg of linoleic acid for women with a high risk of preeclampsia was dramatically effective in reducing rates of pregnancy-induced hypertension (PIH) (8% in treatment vs. 42% in controls [10]) and preeclampsia (9% vs. 37% [11]).

Preeclampsia almost doubles the risk of children being born with autism [12]. Similarly, one retrospective study [13] in China found that calcium supplementation was associated with a decreased risk of ASD (OR = 0.48, CI = 0.28, 0.84).

One study [14] in Trinidad (Pacific Islands) of 510 women found that 1200 mg of calcium was much more effective than 600 mg of calcium plus aspirin at reducing the risk of preeclampsia, PIH, and hypertension in a population where these risks were relatively high (10–25%).

In one study [15] in Hong Kong, pregnant women with a moderate to high risk of PIH were supplemented with placebo or calcium (600 mg calcium from 22 to 32 weeks, and 1200 mg/day from week 32 to birth). Supplementation reduced the rate of proteinuric PIH (more severe PIH) to 5.6% in the calcium group (*n* = 154) vs. 10.7% in the placebo group (*n* = 83), *p* = 0.06. Similarly, the rates of non-proteinuric PIH were 15% and 24%, respectively.

Another study [16] investigated doses of 120 mg, 240 mg, 1000 mg, 2000 mg, or placebo and found that the incidence of pregnancy-induced hypertension was 8.9%, 7.5%, 8%, and 4%, respectively, in these groups vs. 18% in the control group—this suggests that even 120 mg may be beneficial, but 2000 mg may be best for some. In a 2011 report, the World Health Organization (WHO) recommended 1.5–2 g/day supplementation of calcium in areas with low dietary intake of calcium, especially for those at high risk of developing preeclampsia [17].

Overall, low levels of calcium are associated with many maternal and infant health problems, and calcium supplementation has many benefits at moderate and high doses and only a possible rare risk of HELLP at high dosages (1500–2000 mg/day).

#### 3.1.2. Intake

The NHANES study found that from 2017 to 2018, the average daily dietary intake (not including supplements) of calcium of U.S. women aged 20–39 was 872 mg/day [18], which was only 87% of the RDA recommendation of 1000 mg of calcium for pregnant women age 20–39 [19]. The Tolerable Upper Limit of calcium intake is 2500 mg [19].

#### 3.1.3. Recommendation

Calcium levels decrease during pregnancy unless supplemented, so for U.S. women, we recommend that prenatal supplements contain approximately 550 mg of elemental calcium, with some women needing more depending on their diet (i.e., if they have a low intake of milk or milk-based foods or low vegetable intake). For women with higher risks of preeclampsia, preterm birth, gestational hypertension, or who have a calcium-deficient diet, we recommend that they supplement with at least 1000 mg of calcium. This recommendation will likely reduce the risk of preeclampsia and possibly help with other conditions such as preterm birth, gestational hypertension, low birth weight, neonatal mortality, and autism. Note: Vitamin D should be given with calcium to improve calcium absorption and to prevent cellular apoptosis or artery hardening due to excess calcium.

#### 3.1.4. Comparison with Commercial Prenatals

Calcium is included in 78% of prenatal supplements; when included, the median level is 200 mg (Q1: 150/Q3: 300.0). Only 8% of prenatal supplements meet or exceed our recommendation for calcium.

### 3.2. Chromium

#### 3.2.1. Research

Chromium is recommended to control blood sugar levels, and low levels are associated with diabetes. One study [20] measured chromium content in hair twice during pregnancy and found it decreased as the pregnancy progressed. Similarly, two studies [21,22] found that chromium levels in hair were very low in pregnant women (at the ends of their pregnancies) compared with non-pregnant controls, and one of the studies found that it took four years for these levels to return to normal. One study [23] found that diabetic mothers and their infants (*n* = 76) had lower levels of chromium in scalp hair compared to the referent mothers–newly-born infant pairs (*n* = 68), and diabetic mother–-newly born infant pair samples had high urinary excretion of chromium. In contrast, a study [20] found that that women with gestational diabetes had much higher chromium in their hair than nondiabetic pregnant women, and pregnant women with pre-gestational overt diabetes mellitus had intermediate levels. Further research is needed on levels of chromium during pregnancy, but the limited data from hair analysis suggests that chromium levels may decrease during pregnancy.

One study [24] in India found that pregnant women with gestational diabetes had much lower levels of serum chromium. In comparison, another study in the U.S. found no difference in serum chromium in 396 nondiabetic pregnant women vs. 29 pregnant women who had gestational diabetes [25]. The difference is likely due to lower levels of chromium in India [24].

One study [26] in the U.S. provided 4 μg/kg of chromium or placebo to 24 women with gestational diabetes. Supplementation significantly improved their fasting plasma glucose and insulin levels and significantly improved their glucose and insulin response to a 100 g glucose loading test. However, in this study, women with severe glucose intolerance had only a partial improvement and still needed insulin therapy. They hypothesized that higher dosages might be necessary [26].

One study [27] found that eight weeks of supplementation of 200 μg of chromium in non-pregnant women with polycystic ovary syndrome (an insulin-resistant condition) led to several benefits and possibly an increase in pregnancy rate (17% vs. 3%, *p* = 0.08) during that eight weeks. Similarly, another study [28] found that eight weeks of supplementation with 200 μg/day of chromium in infertile women with polycystic ovary syndrome led to significant improvements in biomarkers of inflammation and glucose.

Chromium supplementation is commonly used in treatment of diabetes (improving glycemic control), and although controversial as to whether or not it is beneficial (due in part to some studies using chromium chloride, an ineffective form), a meta-analysis of 22 randomized controlled trials found that it improved glycemic control, reduced triglycerides, and increased HDL-C (high density lipoprotein-cholesterol) levels [29]. Chromium picolinate was the form with the greatest effects on glucose and triglyceride levels [29]. Doses over 200 μg/day were more effective at improving glycemic control, and there was no difference in the adverse events between chromium and placebo [29].

Several studies have suggested that chromium supplementation at 400–600 μg/day is helpful for atypical depression [30]. Therefore, we speculate that it may also help prevent post-partum depression, and research in this area would be beneficial.

Overall, low chromium is associated with gestational diabetes, and chromium supplementation improves glucose and insulin levels.

#### 3.2.2. Intake

There is no NHANES data on the average daily intake of chromium, but it is estimated that adult women consume about 23 to 29 μg of chromium per day from food [19]. The RDA for chromium is 30 μg/day for pregnant women [19]. Chromium is generally well-tolerated [19], and no Tolerable Upper Limit has been established; in 1989, the National Academy of Sciences established an “estimated safe and adequate daily dietary intake” of 50 to 200 μg/day for adults and adolescents.

#### 3.2.3. Recommendation

Limited data in hair suggests that chromium levels decrease during pregnancy. For U.S. women, we recommend that prenatal supplements contain at least 100 μg/day of chromium as chromium picolinate, although more research is needed to determine optimal dosing, and women who develop gestational diabetes or women with diabetes should use 200 μg/day or more to improve glycemic control. The best form is chromium picolinate; the worst form (mostly ineffective) is chromium chloride. This recommendation may reduce the risk of gestational diabetes and possibly improve fertility in women who have PCOS.

#### 3.2.4. Comparison with Commercial Prenatals

Chromium is included in 35% of prenatal supplements; when included, the median level is 79 μg (Q1: 30/Q3: 120). Sixteen percent of prenatal supplements on the market contain enough chromium to meet or exceed our recommendation.

### 3.3. Copper

#### 3.3.1. Research

Copper is needed for several functions, including iron absorption, formation of connective tissue, energy metabolism, oxidative stress, and brain development. Copper is recommended to prevent miscarriages and is necessary to prevent anemia because copper-based enzymes are needed for iron absorption.

Copper levels increase during pregnancy, eventually reaching twice that of non-pregnant women, due to increasing estrogen [31]. One study of over 800 problem pregnancies found that serum copper levels were much lower throughout pregnancy compared to healthy pregnant controls and especially lower in those involving spontaneous abortion of the fetus [32]. Similarly, other studies have found lower copper in women with spontaneous abortion, threatened abortion, missed abortion and blighted ovum [31,33], and in anencephalic pregnancies [33]. Another study found that in drinking water, copper was the mineral most strongly associated with central nervous system (CNS) malformations [34]. One study found that low copper was associated with premature rupture of membranes (PROM) in preterm, but not term, pregnancies [35]. A meta-analysis of 17 studies of Asian populations [36] and a meta-analysis of 12 Asian and European studies [37] found that copper levels were higher during the third trimester in women with preeclampsia, and several studies have reported increased ceruloplasmin during pregnancy, and 96% of serum copper is bound to ceruloplasmin—therefore, increased oxidative stress associated with preeclampsia may result in an increased production of ceruloplasmin (an anti-oxidant) to combat the preeclampsia. It is unclear if copper is causing oxidative stress (as free copper) and an increased risk of preeclampsia, or if it is merely a symptom of the body’s anti-oxidant response to preeclampsia. A study in France provided 2 mg of copper as part of a multi-vitamin/mineral supplement, and this resulted in no difference in copper levels between treatment and placebo groups [38].

One RCT involving 1 mg/day of copper found that supplementation resulted in significant decreases in depression (75% and 90% decreases in the second trimester and third trimester, respectively) and significant decreases in anxiety (45% and 80% decreases in anxiety symptoms in the second trimester and third trimester, respectively) [39]. There was also a significant decrease in rate of infection during pregnancy (*p* = 0.046).

Overall, it appears that low copper in early pregnancy is a risk factor for spontaneous abortion and CNS malformations, so supplementation before conception seems essential, and low copper in later pregnancy is a risk factor for PROM.

#### 3.3.2. Intake

The NHANES study found that from 2017 to 2018, the average daily dietary intake (not including supplements) of copper of U.S. women aged 20–39 was 1.1 mg/day [18], which was slightly more than the RDA recommendation of 1 mg/day for pregnant women [19]. The Tolerable Upper Limit for pregnant women is 10 mg/day [19].

#### 3.3.3. Recommendation

Therefore, for U.S. women we recommend that prenatal supplements contain approximately 1.3 mg of copper, based on the limited research to date. Supplementation before conception may help decrease the rate of miscarriages and CNS defects, and supplementation at that time or later may reduce the risk of anemia and PROM.

#### 3.3.4. Comparison with Commercial Prenatals

Copper is included in 58% of prenatal supplements; when included, the median level is 1.5 mg (Q1: 1.0/Q3: 2.0). Forty-one percent of prenatal supplements on the market meet or exceed our recommendation.

### 3.4. Iodine

#### 3.4.1. Research

Low iodine is strongly associated with iodine deficiency disorders, including hypothyroidism in mothers and infants and intellectual disability in infants. Low iodine is possibly related to preeclampsia [40] and autism [41,42]. Low iodine results in reduced thyroid hormone synthesis, which causes increased pituitary TSH production, which stimulates thyroid growth and can result in maternal or fetal goiter.

Iodine supplementation is strongly recommended for pregnant women to decrease the rate of hypothyroidism and intellectual disability in their children [43]. Worldwide, 2 billion people have insufficient iodine intake, 31% of children have insufficient iodine intake, and, hence, it is a common cause of hypothyroidism, intellectual disability, and intellectual impairment [44]. In the U.S., iodine deficiency was especially common in the iodine-deficient regions of the Great Lakes, Appalachians, and the northwestern area of the country, affecting 26–70% of the population in those regions until the 1920s, when the use of iodized salt began [45]. Worldwide, 120 countries now require iodization of salt, but in the U.S., it is still optional, only 70–76% of the U.S. population uses iodized salt [45], and half the salt brands in the U.S. use less than the level recommended by the FDA. Furthermore, from the 1970s to the 1990s, there was a 50% decrease in median urinary iodine levels in the U.S. [46], apparently due to decreased use of iodized table salt, so an increasing number of women and infants are at risk of iodine deficiency. The median urinary iodine concentration in women aged 15–44 years in the U.S. in the 2011–2014 National Health and Nutrition Examination Survey (NHANES) [47,48] surveys was 110 mcg/L, which was mildly deficient compared to World Health Organization (WHO) [49] guidelines of 149–249 μg/L for optimal levels and 50–150 for marginally deficient. Iodine requirements increase during pregnancy due to at least three factors: (1) increased need for thyroxine (T4) to maintain normal metabolism in the mother, (2) a transfer of iodide and T4 from the mother to the fetus, and (3) possible increased loss of iodide from the kidneys [50]. Iodine levels may remain stable during pregnancy if sufficient iodine is present in the diet [51], but in marginally-deficient regions, iodine levels decrease during pregnancy [52,53]. An extensive review of iodine status recommended that the optimal range for total iodine intake during pregnancy was in the range of 250–300 μg [54].

Neurodevelopmental delay is increased in children born to mothers with iodine deficiency [55]. Iodine deficiency needs to be corrected very early in pregnancy, as low levels after mid-gestation result in permanent damage. Higher psychomotor test scores were observed in children whose mothers were supplemented with iodine before the third trimester in comparison to those who were provided iodine later in pregnancy or at 2 years of age [56].

Severe iodine deficiency during pregnancy results in increased rates of miscarriage, stillbirth and infant mortality, and impaired cognitive function of infants [57]. It can also result in cretinism, which involves severe intellectual disability, deaf–mutism, and motor rigidity. Iodine deficiency is the leading cause of preventable intellectual disability worldwide.

Mild to moderate iodine deficiency during pregnancy increases the risk of maternal goiter and thyroid disorders [58] as well as attention deficit disorder [59] and impaired cognition in the infants [60,61,62]. A meta-analysis [63] of three studies involving 9036 mother–child pairs in Europe found that low maternal free thyroxin (T4, the major thyroid hormone produced by the thyroid gland) in early pregnancy was associated with a lower IQ in all three studies; T4 below the 2.5th percentile was associated with a 3.9-point lower nonverbal IQ and a 2.1-point lower verbal IQ.

The effect of iodine supplementation during pregnancy clearly reduces maternal thyroid disorders, but results are mixed on whether or not it can improve neurodevelopment [64]. A systematic review and meta-analysis [65] of 37 studies found that iodine supplementation during pregnancy reduced maternal thyroglobulin, and 3 RCTs found it reduced the increase in maternal thyroid volume during pregnancy. However, the review found that it did not affect maternal or infant thyroid-stimulating hormone or free thyroxine, and limited studies found it did not improve intellectual development of the infant. So, it appears that iodine supplementation may need to be given preconception in order to improve infant intellectual development.

A systematic review [66] of 28 studies found “moderate evidence for an association between maternal thyroid hormone levels and offspring ADHD (attention deficit hyperactivity disorder).” Three of four studies found an association of maternal hypothyroidism and ADHD. Maternal hypothyroxinemia (normal TSH (thyroid stimulating hormone) levels and low free thyroxin (fT4)) was associated with an increased risk of ADHD in children (OR = 1.54 (CI = 1.02, 2.33), 4 studies).

One study found lower levels of iodine in women with severe preeclampsia [40]. A small, randomized, double-blind, placebo-controlled study [67] in a region of Italy with mild/moderate iodine deficiency found that supplementation with 225 μg/day of iodine starting during the first trimester was not detrimental, greatly increased urinary iodine levels from moderate deficiency to a normal range, and reduced thyroid hyperstimulation. One small, open-label study in a region of Italy with mild iodine deficiency found that supplementing with 200 μg/day resulted in slightly better improvements in thyroid function than supplementing with 50 μg/day [68].

One study found lower levels of iodine in children with autism [41], and another study found 25% of children with autism had unusually low levels of iodine [41], suggesting a possible link between iodine deficiency and the risk of autism, consistent with studies that iodine deficiency affects language skills [69]. The World Health Organization, UNICEF (United Nations Children’s Fund), and ICCIDD (International Council for Control of Iodine Deficiency Disorders) [49] recommend that pregnant women be supplemented with 250 μg/day of iodine in countries with insufficient access to iodized salt to prevent iodine deficiency disorders including hypothyroidism and goiters in infants. The American Thyroid Association taskforce recommended 150 mg/day supplementation for pregnant women, starting 3 months prior to pregnancy [53].

Chronic excessive iodine intake can inhibit the production of thyroid hormones (known as the Wolff–Chaikoff effect) [70]. A detailed study of 7190 pregnant women in an iodine-sufficient region of China found that women with UIC (Urinary Iodine Concentration) of 150–249 μg/L had the lowest rate of subclinical hypothyroidism and isolated hypothyroxinemia, UIC levels of 250–499 μg/L resulted in a 1.7-fold increase of subclinical hypothyroidism, and UIC levels above 500 μg/L were associated with a 2.8-fold increase of hypothyroxinemia [71].

Similarly, severe excess maternal intake of iodine (800–12,000 ug/day) can cause thyroid disorders, including congenital hypothyroidism in infants [72,73,74], and although effects are often transitory, some children require levo-thyroxine due to hypothyroxinemia or persistent hyperthyrotropinemia. Such high levels are uncommon in the U.S., and in a study of 907 cases of congenital hypothyroidism and 900 controls in California (U.S.), there were no differences in median blood iodine levels nor in the distribution of it at the high or low ends, suggesting that iodine levels at birth were not a major contributing factor [75]. Therefore, although the Tolerable Upper Intake is set at 1100 μg/day for all people, including pregnant women, the WHO [49] has recommended that total daily intake above 500 μg/day may be excessive.

#### 3.4.2. Intake

The NHANES study found that from 2011 to 2014, the average daily urinary iodine concentration of U.S. women aged 15–44 years was 110 μg/L, which corresponded to a daily intake of approximately 160 μg/day [49] based on standard assumptions of 100% absorption of dietary iodine, 90% excretion in the urine, and urinary excretion of 1400 mL/day [76], This is lower than the RDA recommendation of 220 μg of iodine for pregnant women [19]. Recent preferences for reduced use of salt and the use of sea salts over iodized table salt have dramatically reduced the amount of iodine in the diet. The Tolerable Upper Intake is 1100 μg [19].

#### 3.4.3. Recommendation

Iodine requirements increase during pregnancy, so for U.S. women, we recommend that prenatal supplements contain approximately 150 μg of iodine, with some women needing more or less depending on their intake of iodized salt and dairy (a major dietary source of iodine) [48]. This recommendation appears likely to reduce the current rate of hypothyroidism and intellectual disability in the U.S. and possibly help with other conditions, as well.

#### 3.4.4. Comparison with Commercial Prenatals

Iodine is included in 76% of prenatal supplements; when included, the median level is 150 μg (Q1: 150/Q3: 225). Sixty-nine percent of prenatal supplements on the market meet or exceed our recommendation for iodine.

### 3.5. Iron

#### 3.5.1. Research

Iron is primarily needed for hemoglobin to transport oxygen in red blood cells, as well as several other roles. Two large studies [77,78] found that about half of U.S. women had indicators of low iron. Similarly, iron deficiency is common in U.S. women, affecting 8.8% of non-pregnant women ages 20–35 and 11.6% of non-pregnant women ages 35–49 [79]. Iron levels decline during pregnancy, so rates of iron deficiency increase during pregnancy, reaching approximately 27.5% by the third trimester [79]. Rates of iron deficiency are lower for the first pregnancy but increase with increasing pregnancies [79]. Low iron levels cause anemia (low hemoglobin), which causes weakness, fatigue, reduced cognitive performance, and diminished immune response in the mother and substantially impairs brain development of the fetus, which can have life-long effects on intellectual ability.

One study found that women with low iron had a doubled risk of having a child with autism [80], presumably due to the critical need for oxygen transport to the brain, the highest oxygen-consuming organ in the body. One study found that the risk of orofacial clefts was significantly lower in mothers with dietary intakes of 14–22 mg/day of iron (OR 0.4) [81]. One study found that dietary intake in the highest quartile (above 18 mg/day) was associated with a significantly decreased risk of anencephaly, with an OR of 0.51 (95% CI: 0.28, 0.96) [82].

Iron supplementation decreases the risk of anemia in mothers and infants [83,84,85,86,87,88]. Three iron supplementation studies in the U.S. [77,78,84] investigated the effects of iron (30–60 mg) or placebo on women with initially normal iron status, supplementing them for approximately weeks 20–28, followed by iron (30–60 mg) for all from weeks 28 on. Iron supplementation was generally beneficial, but there was still a dramatic decrease in serum ferritin and a substantial decrease in hemoglobin despite supplementation of 30–60 mg. One of the studies [84] used much higher doses of 60 mg 3×/day (180 mg total) for 12 participants with iron deficiency anemia (IDA), and that reduced IDA in 75% of the subjects. Therefore, it appears that women with initially normal iron status still need about 60 mg starting at week 20, and some may need up to 180 mg or more.

A meta-analysis of 44 trials involving 43,274 women compared the effects of daily iron supplements (mostly 30–200 mg/day) versus no iron or placebo [89]. Preventive iron supplementation reduced maternal anemia at term by 70% (14 trials, 2199 women), iron-deficiency anemia at term by 67% (6 trials, 1088 women), and iron deficiency at term by 57% (7 trials, 1256 women, low quality evidence). Overall, these studies showed that iron supplementation was helpful, but that some studies used too low a level of supplementation to fully eliminate iron-deficient anemia.

#### 3.5.2. Intake

The NHANES study found that from 2017 to 2018, the average daily dietary intake (not including supplements) of iron by U.S. women aged 20–39 was 12.2mg/day (U.S. Department of Agriculture, 2014), which was only half of the RDA recommendation of 27 mg iron for pregnant women [19]. The Tolerable Upper Limit is 45 mg [19], but many studies have found benefits from doses of 60 mg during pregnancy.

#### 3.5.3. Recommendation

Iron levels decrease substantially during pregnancy, and some women begin pregnancy with iron deficiency. Therefore, for U.S. women, we recommend that prenatal supplements contain at least 30 mg of iron during the first trimester and at least 60 mg starting week 20, with some women needing up to 60 mg 3×/day if iron levels are low or iron deficiency anemia occurs. We strongly recommend testing iron levels before conception, or as soon after as possible, and re-testing at the start of the third trimester, when the need for iron increases substantially. Note that iron deficiency is extremely common in women, and fetal brain development during the first months is especially sensitive to iron deficiency, so it is important to normalize iron levels as soon as possible. Women with a history of anemia and with multiple pregnancies are especially at risk. This recommendation appears likely to reduce the rate of anemia and intellectual disability in the U.S. and possibly help decrease the risks of autism and anencephaly. Note that copper is needed for copper-based enzymes that absorb iron, so low copper can cause low iron.

#### 3.5.4. Comparison with Commercial Prenatals

Iron is included in 89% of prenatal supplements; when included, the median level is 27 mg (Q1: 19.5/Q3: 28). Seventeen percent of prenatal supplements on the market contain iron at levels that meet or exceed our recommendation.

### 3.6. Magnesium

#### 3.6.1. Research

Magnesium is needed as an enzymatic co-factor for hundreds of reactions in the body and has many functions including cell signaling, ion transport, energy production, and synthesis of nucleic acids and proteins. It is also an important part of bones, cell membranes, and chromosomes.

Blood levels of magnesium decrease significantly during pregnancy [90,91,92,93], but supplementation with 345 mg/day is enough to keep magnesium levels stable [94]. Low magnesium is strongly associated with increased risk of preeclampsia [95,96,97], preterm labor [97,98], and leg cramps [97] and slightly associated with pregnancy-induced hypertension [7]. One study found that the risk of orofacial clefts was significantly lower in mothers with dietary intakes of 376–404 mg/day of magnesium (OR 0.4) [81]. One ecological study across the U.S. found a strong inverse correlation between magnesium levels in soil and the rates of preterm birth (*r* = −0.68; *p* < 0.001) [99].

A meta-analysis of magnesium supplementation with 345 to 500 mg/day (Makrides [100]) with re-analysis by the present authors to remove one study with a very low dosage averaging 32 mg/day [101]) found significant decreases in the rate of pregnancy-induced hypertension [102,103] as well as reduced maternal hospitalization during pregnancy [94,104,105], preterm birth [104,105,106,107], low birth weight [104,105,106], and babies with low Apgar scores (4 studies, review by Makrides [100]). One study found a dosage of 184 mg/day of magnesium gluconate substantially reduced the rate of pregnancy-induced hypertension [103]. One study used a dosage averaging only 32 mg/day and found little benefit [101]. Overall, magnesium supplementation was well-tolerated, and doses of 184–500 mg/day resulted in multiple benefits in all but one study.

Magnesium supplementation resulted in a very large reduction in the rate of pregnancy-induced hypertension in the two studies in Angola [102] and China [103]; the combined rate was 4% in the treatment group vs. 21% in the placebo group (out of a total of 201 participants). The Angola study also reported a substantial reduction in the rate of edema (24% vs. 58%, respectively). However, a study of magnesium supplementation (300 mg/day of elemental magnesium) in Brazil [108] did not find any effect on perinatal composite outcome (preterm birth, stillbirth, neonatal death, NICU admission, or small-for- gestational-age birthweight) or for maternal composite outcome (preeclampsia, eclampsia, gestational hypertension <37 weeks, placental abruption, or maternal stroke or death during pregnancy or ≤7 days after delivery). Therefore, the effects of magnesium supplementation may vary between countries.

A more recent metanalysis of six RCT’s with 3068 pregnant women found that magnesium supplementation significantly reduced the rate of preterm birth (RR = 0.58, CI = 0.35–0.96) [99].

Several studies investigated giving high doses of magnesium sulfate (4–6 g) intravenously to mothers at high risk of preterm birth (birth expected within 24 h, gestational age less than 33 weeks), to reduce the risk of cerebral palsy [109]. A meta-analysis of five studies found that magnesium sulfate significantly reduced the risk of cerebral palsy by approximately 32% and significantly improved the rate of gross motor dysfunction [109]. There were frequent minor side effects to the mothers (flushing, sweating, nausea, vomiting, headaches, and a rapid heartbeat (palpitations)), but no major complications.

#### 3.6.2. Intake

The NHANES study found that from 2017 to 2018, the average daily dietary intake (not including supplements) of magnesium of U.S. women aged 20–39 was 269 mg/day [18], which was 22% less than the RDA recommendation of magnesium for pregnant women of 350–360 mg for women ages 19 to 50 [19]. The official Tolerable Upper Limit of magnesium in a supplement is 400 mg [19], but eight treatment studies using levels of 345–500 mg were generally very beneficial [100], so the Tolerable Upper Limit evidently needs to be updated for pregnant women.

#### 3.6.3. Recommendation

Overall, because U.S. women have average magnesium intake that is 22% less than the RDA, and because levels decrease during pregnancy, we recommend supplementing with 350 mg because 345 mg was found be sufficient to keep magnesium levels stable, and supplementation studies with doses of 345–500 mg were found to be beneficial. We recommend avoiding the oxide form due to poor absorption. This recommendation appears likely to reduce the rate of pregnancy-induced hypertension, maternal hospitalization, preterm birth, low birth weight, and low Apgar scores, and could possibly help with other conditions as well.

#### 3.6.4. Comparison with Commercial Prenatals

Magnesium is included in 66% of prenatal supplements; when included, the median level is 50 mg (Q1: 40/Q3: 181). Only 5% of prenatal supplements on the market meet or exceed our recommendation for magnesium.

### 3.7. Manganese

#### 3.7.1. Research

Manganese is vital for healthy brain and nervous system function as well as maintaining metabolism and hormone production. Manganese is one of the least-studied micronutrients for pregnancy, and at present, no supplementation trial for pregnancy has been published. Manganese is an essential mineral, but an excessive amount can cause neurological disorders. Manganese levels appear to increase throughout pregnancy [110] due to low iron levels, leading to increased absorption of manganese. Low levels of manganese are associated with lower birth weight [111,112], fetal growth restriction [111], and possibly with preterm labor [113]. One study in Korea [114] found that the mean manganese concentration in whole maternal blood was 22.5 μg/L, and birth weight peaked at the maternal blood manganese levels of 30 and 35 μg/L and was lower if manganese was low or unusually high. One study [115] did not find an association of blood levels of manganese with fetal growth, but the mothers were all exposed to manganese-containing pesticides, which may have confounded the results. One study found that mothers with preeclampsia had higher blood levels of manganese [116]. One study in Korea found that low or high levels of manganese were associated with worse mental and psychomotor development at six months of age, and that median levels were best [117]. One study found that cord blood manganese above the 75th percentile was negatively associated with neurodevelopment scores at age two years, including worse overall neurodevelopment (β = −7.03, *p* = 0.009) and cognitive (β = −8.19, *p* = 0.01), and language quotients (β = −6.81, *p* = 0.01) [118]. One large study of 397 children with ASD and 1034 controls found that maternal blood levels in the highest quartile were associated with an increased risk of ASD (OR = 1.84) (CI: 1.30, 2.59) [119]. Overall, it appears that low levels or unusually high levels of manganese are a problem for fetal growth, neurodevelopment, and psychomotor development.

#### 3.7.2. Intake

Based on the Total Diet Study, the average intake of manganese was 2.3 mg/day for women age 19–34, which was slightly more than the RDA recommendation of 2.0 mg for pregnant women [19]. The Tolerable Upper Limit is 11 mg/day [19].

#### 3.7.3. Recommendation

For U.S. women, we recommend that prenatal supplements contain approximately 1 mg of manganese, although more research is needed to improve this recommendation. When iron is supplemented, manganese absorption will decrease; therefore, manganese supplementation is more likely to be needed. This recommendation appears likely to reduce the risk of low birth weight, fetal growth restriction, and possibly preterm labor; moderate levels of manganese are associated with improved mental and psychomotor development.

#### 3.7.4. Comparison with Commercial Prenatals

Manganese is included in 40% of prenatal supplements; when included, the median level is 2.0 mg (Q1: 2.0/Q2: 3.9). Thirty-eight percent of prenatal supplements on the market meet or exceed our recommendation for manganese.

### 3.8. Molybdenum

#### 3.8.1. Research

Molybdenum is an essential co-factor for three enzymes. Molybdenum deficiency is rare, but one study found that about 38% of children with autism often needed 50 μg molybdenum to normalize activity of one of the enzymes to improve sulfation [120]. There is almost no data on the optimal level of molybdenum supplementation during pregnancy. One study found an association between above-average levels of molybdenum with slightly worse psychomotor development in infants [121], but that was likely due to exposure to toxic forms of molybdenum and not the forms found in food.

#### 3.8.2. Intake

The Total Diet Study estimated daily intake of 76 μg/day for women [122], which was more than the RDA recommendation of 50 μg/day for pregnant women [19]. The Tolerable Upper Limit for pregnant women is 2000 μg [19].

#### 3.8.3. Recommendation

We tentatively recommend 25 μg/day to prevent molybdenum deficiency in women with low dietary intake; more research is needed to determine the optimal dosage and possible health benefits.

#### 3.8.4. Comparison with Commercial Prenatals

Molybdenum is included in 29% of prenatal supplements; when included, the median level is 25 μg (Q1: 50/Q3: 75). Twenty-eight percent of prenatal supplements on the market meet or exceed this recommendation.

### 3.9. Selenium

#### 3.9.1. Research

Selenium has many functions in the body, primarily as selenocysteine-containing proteins (selenoproteins). Selenium supplementation has been shown to reduce hypothyroidism, pregnancy-induced hypertension, and preeclampsia, as well as postpartum depression, in pregnant women. It has been found that selenium stores in the body are depleted throughout pregnancy, with most depletion occurring at the end of pregnancy [123,124,125,126]. A meta-analysis of 7 studies found that serum selenium levels were lower in women with gestational diabetes mellitus (GDM) than those with standard glucose tolerance, especially in Asia, but not in Europe [127]. A review [128] of selenium status and adverse pregnancy outcomes found that low selenium was possibly associated with miscarriages (2 studies, and mixed results for a 3rd), neural tube defects (3 of 4 studies), low birth weight (5 of 8 studies), intrauterine growth restriction (1 of 3 studies), congenital diaphragmatic hernia (1 large study), preeclampsia (3 of 6 studies), and abnormal glucose metabolism and gestational diabetes (3 studies). One study found that low selenium levels during pregnancy were associated with a higher risk of infection during the first 6 weeks of life and a worse psychomotor score at 6 months [126]. Selenium supplementation in mothers results in a higher selenium status in their infants [129].

A study of 230 women in the UK found that 60 μg/day of selenium was enough to prevent a decrease of selenium levels, improve the primary biomarkers of selenium status, and significantly reduce the odds of having either preeclampsia or PIH (65% reduction, *p* = 0.04) [124]. A randomized trial [130] of 169 women compared 100 μg/day of selenomethionine with placebo, and no significant differences were seen for either preeclampsia or preterm birth, although the study was probably too small to detect significant differences. In women at risk of pregnancy-induced hypertension, a dose of 100 μg during the last 6–8 weeks of pregnancy significantly reduced the risk of pregnancy-induced hypertension (8% vs. 23%, *p* < 0.05) and gestational edema (15% vs. 33%, *p* < 0.05) and cured existing gestational edema in 88% of 26 cases vs. 1 out of 13 cases in the placebo group, *p* < 0.0001 [130]. A RCT of supplementation of 200 μg/day of selenium for 6 weeks in 70 women with gestational diabetes found significant improvements in fasting glucose, serum insulin, C-reactive protein, and oxidative stress [131].

Selenium has also been shown to treat postpartum depression when taken at levels of 100 μg/day [130]. In women at high risk of postpartum thyroid deficiency, supplementation with 200 μg/day greatly reduced the rate of postpartum thyroid deficiency (29% vs. 49%, *p* <0.01) and permanent hypothyroidism (12% vs. 20%, *p* < 0.01) [132].

#### 3.9.2. Intake

The NHANES study found that from 2017 to 2018, the average daily intake (not including supplements) of selenium of U.S. women aged 20–39 was 97 μg/day [18], which was more than the RDA recommendation of 60 μg/day of selenium for pregnant women [19]. The Tolerable Upper Limit is 400 μg/day [19].

#### 3.9.3. Recommendation

Because selenium levels decrease during pregnancy and because one study found 60 μg/day was sufficient to prevent that decrease, we recommend 60 μg/day, preferably as selenomethionine. For women with gestational edema or pregnancy-induced hypertension, 100 mg is recommended. For women at risk of postpartum thyroid deficiency, 200 μg/day or higher is recommended. This recommendation appears likely to reduce the risk of postpartum depression, gestational diabetes, pregnancy-induced hypertension/preeclampsia, and postpartum thyroid deficiency as well as possibly neural tube defects, miscarriages, intrauterine growth restriction, and hernias.

#### 3.9.4. Comparison with Commercial Prenatals

Selenium is included in 40% of prenatal supplements; when included, the median level is 70 μg (Q1: 35/Q3: 70). Twenty-four percent of prenatal supplements meet or exceed our recommendation.

### 3.10. Zinc

#### 3.10.1. Research

Zinc has many roles in the body, including immune function, growth and development, nerve function, vision, and fertility. Zinc is recommended for reducing the risk of preeclampsia in pregnant women and reducing the risk of preterm birth as well as asthma in infants. Zinc absorption increases during late pregnancy and early lactation [133], but most studies suggest serum zinc levels decrease somewhat throughout pregnancy [134,135,136,137,138], although one study in China found a small increase in levels [139]. By the end of pregnancy, levels were significantly lower in comparison to healthy, non-pregnant controls in most countries (20% lower in Turkey [140], 38% lower in Korea [125], 38% lower in U.S. [136], but 10% higher in China [139]—in China, zinc levels were much higher than in other countries.

Two meta-analyses, both including 14 studies, confirmed low serum zinc levels were associated with PIH and preeclampsia [7,141].

A meta-analysis of 32 studies of maternal dietary intake found that higher maternal intake of zinc was associated with a lower probability of wheeze (OR = 0.62, 95% CI = 0.40–0.97) during childhood (but not necessarily asthma) [142]. Zinc levels in mothers were negatively correlated with asthma in their children at the age of five [143].

A study of 450 pregnancies [144] included an evaluation of 12 biomarkers of nutritional status and found that zinc levels in the lowest quartile had the strongest association with total occurrence of fetal and maternal health complications and the specific symptom of fetal distress.

One study [145] of 279 pregnant women found that plasma zinc levels at delivery below the median level were associated with “mild toxemia (5.6% vs. 0.7%, *p* = 0.02), vaginitis (13% vs. 4.4%, *p* = 0.01), and postdates (4.2% vs. 0%, *p* = 0.01) in the antenatal period. During the intrapartum period, low plasma zinc levels were associated with a prolonged latent phase (2.8% vs. 0%, *p* = 0.05), a protracted active phase (29% vs. 18%, *p* = 0.04), labor greater than 20 h (6.3% vs. 1.5%, *p* = 0.03), second stage greater than 2.5 h (6.3% vs. 0.7%, *p* = 0.01), and cervical and vaginal lacerations (7.0% vs. 1.5%, *p* = 0.02)”. No such association was found with red blood cell zinc, suggesting that plasma zinc was a better biomarker.

Regarding supplementation effects, one study in the U.S. with women with initial plasma zinc levels below the median value found that by comparing levels at the beginning and the end of pregnancy, zinc levels had decreased by 11% in the un-supplemented group vs. only 7% in a group receiving 25 mg/day of zinc sulfate [135]. One U.S. study [136] of healthy, middle-income Caucasians found plasma zinc levels decreased during pregnancy, and supplementation averaging 11 mg/day of zinc sulfate had little effect on plasma levels but did increase hair and possibly saliva levels. One study [134] of U.S. Hispanic women found that serum zinc declined during the first trimester, and that levels remained similarly low in those taking or not taking 20 mg/day of zinc acetate, although there was a significant decrease in the percent with deficient levels (20% decreased to 10%, *p* < 0.05) in the treatment group and not in the unsupplemented group. In another U.S. study [146], 138 low-income Hispanic teenagers were randomized to a vitamin/mineral supplement with or without 20 mg of zinc sulfate. Serum zinc levels decreased 14% in the non-supplemented group vs. 8% in the supplemented group from about the 17th week to week 31–36 of pregnancy [146]. Zinc supplementation reduced (*p* = 0.018) the number of low serum zinc values (less than or equal to 53 micrograms/dl) in late pregnancy [146]. Zinc supplementation did not affect the outcome of pregnancy, but serum zinc levels were lower (*p* = 0.038) in teenagers with pregnancy-induced hypertension than in normotensives [146].

A meta-analysis of 16 randomized, controlled trials found that zinc supplementation resulted in a 14% reduction in preterm birth (RR = 0.86, CI = 0.76 to 0.97), but no other significant benefits were found [147]; most of the studies used 20–30 mg of zinc, with most studies using zinc sulfate. Similarly, another meta-analysis of 12 studies found that zinc supplementation did not affect low birthweight or pre-eclampsia/eclampsia [148].

Daily zinc supplementation of 25 mg/day zinc sulfate in women with relatively low plasma zinc concentrations in early pregnancy was associated with greater infant birth weights and head circumferences [135]; however, zinc supplementation did not affect test scores of neurologic development [149]. In another study, mothers were supplemented with 30 mg/day, which resulted in a 54% reduction in the chance of their infant having impetigo [150], a bacterial skin infection which affects about 162 million children worldwide [151].

One study [138] compared doses of 0, 5, 10, and 30 mg of zinc lactate. The results demonstrated that the doses of 5 and 10 mg had no benefit, but the 30 mg/day dose resulted in great improvements in the incidence of low birth weight, preterm birth, and intrauterine growth retardation [138]. There was also improvement in birth weight, head circumference, duration of gestation, and Apgar scores [138]. Serum zinc decreased in the group receiving placebo, 5, or 10 mg, but 30 mg of zinc lactate was enough to slightly increase zinc levels [138]. More research is needed, but this study suggests that zinc lactate may be more beneficial than other forms of zinc.

Zinc sulfate was most commonly used and likely beneficial at preventing preterm birth; limited data suggest that zinc lactate and zinc gluconate are also beneficial [147]. Two studies suggest that zinc acetate is not beneficial [147].

Overall, it appeared that 30 mg/day of zinc lactate was enough to stabilize zinc levels [138]; there was only a partial benefit of 20 mg/day of zinc sulfate [146] or 25 mg/day of zinc sulfate [135]; there was little effect of 11 mg/day of zinc sulfate [136] or 20 mg/day of zinc acetate [134] on zinc levels.

#### 3.10.2. Intake

The NHANES study found that from 2017 to 2018, the average daily dietary intake (not including supplements) of zinc of U.S. women aged 20–39 was 9.4 mg/day [18], which was somewhat less than the RDA recommendation of 11 mg/day for pregnant women [19]. The Tolerable Upper Limit is 40 mg/day [19].

#### 3.10.3. Recommendation

Therefore, for U.S. women, we recommend that prenatal supplements contain at least 30 mg of zinc, but further research may suggest that higher levels are beneficial. This recommendation appears likely to reduce the risk of preterm birth and may reduce the risk of impetigo, asthma, and preeclampsia. We recommend using zinc sulfate, zinc gluconate, or zinc lactate but avoiding zinc acetate. One study suggests that zinc lactate may be especially beneficial, but more research is required for verification.

#### 3.10.4. Comparison with Commercial Prenatals

Zinc is included in 89% of prenatal supplements; when included, the median level is 15.0 mg (Q1: 11.0/Q3: 20.0). No prenatal supplement on the market meets or exceeds our recommendation.

## 4. Discussion

Essential minerals are vital for human health, and during pregnancy, they are even more critical to support maternal health and infant development. If not supplemented, the levels of many essential minerals decrease during pregnancy, including calcium, iron, magnesium, selenium, zinc, and possibly chromium and iodine. Low levels of minerals are associated with a wide range of maternal and infant health problems that occur in the U.S. and worldwide, and appropriate supplementation can reduce the risk of many maternal and infant health complications.

### 4.1. Associations between Health Outcomes and Mineral Status

Table 1 and Table 2 summarize the associations between health outcomes (maternal and infant, respectively) and mineral status. The associations are listed as “significant” if there are two or more statistically significant studies in support of the association and a ratio of 2:1 or higher of positive to null studies. Associations are listed as “possible” if there is at least one study with a statistically significant association, and the ratio of positive to negative studies is greater than or equal to 1:1. In cases where meta-analyses were performed, the results of the meta-analysis were counted as “significant” evidence if the results were statistically significant. In general, Table 1 and Table 2 indicate that many maternal and infant health complications are associated with one or more minerals.

Table 3 and Table 4 list the same data as in Table 1 and Table 2, but organized by minerals; i.e., Table 3 and Table 4 list associations between maternal mineral status and maternal and infant health complications, respectively. Overall, it is clear that each mineral is linked to one or more maternal and infant health complications, with the possible exception of molybdenum, which has not been well-researched. It is likely that further research may reveal additional linkages.

Table 5 lists our proposed evidence-based recommendations for prenatal supplements and also lists the RDA, the average daily intake for women ages 20–39 years in the U.S. based only on food/beverages (does not include intake from nutritional supplements), the Tolerable Upper Limit, and if the mineral levels decrease during pregnancy. Comparing the NHANES with the RDA shows that average daily intake of iron, iodine, and magnesium in the U.S. are well below the RDA. Also, it is important to realize that the NHANES values are averages, and some women consume less. So, our recommendations for a general prenatal supplement are intended to help all women achieve at least the RDA without providing an excessive amount that would approach the Tolerable Upper Limit. In addition, levels of many minerals decrease significantly during pregnancy without supplementation, and some of the research that we have summarized in this paper suggests that the RDA may be too low for several minerals in pregnant women, and that higher levels of supplementation may provide improved health outcomes for mother and/or child.

Low levels of essential minerals can affect many pregnancy and infant health complications, as shown in Table 1, Table 2, Table 3 and Table 4, and many of these conditions are common in the U.S. The evidence-based supplement recommendations made in this paper (summarized in Table 5) are likely to significantly reduce many pregnancy complications and infant health problems.

It is important to note that nutritional needs are not static but can change as the pregnancy progresses. Levels of many minerals decrease significantly during pregnancy, including calcium, iron, magnesium, selenium, zinc, and possibly chromium and iodine. These substantial decreases during pregnancy are not all widely known, and strongly argue for increased supplementation of those minerals. For example, many women start their pregnancy with low iron levels, and iron levels decrease further during pregnancy, especially mid-trimester, because of the production of new red blood cells for placenta and fetus. Although iodine levels appear to be stable during pregnancy, low levels of iodine are common in the U.S., so iodine supplementation is also important.

Table 6 reports the mineral content of 188 prenatal supplements and a comparison against our evidence-based recommendations. As shown in Figure 1, the rates of inclusion of each mineral in prenatal supplements were iron (89%), zinc (89%), calcium (78%), iodine (76%), magnesium (66%), copper (58%), manganese (40%), selenium (40%), chromium (35%), and molybdenum (29%). So, although most supplements contained some iron, zinc, and calcium, other essential minerals were included at a lower frequency. When minerals were included in prenatals, the levels were usually below our recommended levels, especially for magnesium, calcium, and zinc, which were included at only 32%, 48%, and 50%, respectively, of our recommendations, if they were included. As shown in Figure 2, although supplements may have included levels of minerals, most included dosages below our recommendations, especially for iron, chromium, magnesium, and zinc, wherein only 17%, 16%, 5%, and 0%, respectively, met our recommendations.

Therefore, analysis of 188 prenatal supplements found that few reached the levels recommended in this review. Overall, the largest differences between our recommendations and the amounts included in prenatals were for magnesium, chromium, and calcium, wherein the average in prenatals was only 21–37% of our recommendations.

Our review of 188 prenatal supplements found that 44% involved only a single capsule/pill/tablet/softgel/gummie. Calcium and magnesium require larger amounts than other minerals, and it appears that manufacturers often limit the amount of those minerals in prenatal supplements so that fewer pills, or only a single pill, are required. Supplements of only 1 pill met only 37% of our recommendations when averaging over all pills, whereas supplements with 6–8 pills met 80% of our recommendations. We estimate that approximately 8 pills, or 2–3 pills 3×/day, would be required to meet our recommendations for vitamins and minerals. This may seem like a large amount, but it is a tiny fraction of the volume of food consumed in a typical diet, and the potential benefits appear to be very high. Today, over 50% of pregnancies involve major complications, including miscarriage (15–20%), iron deficiency (28% by third trimester), and pre-term birth (10%); taking a few extra pills may result in a substantial decrease in those rates, and substantially improve the health of the mother and her baby.

### 4.2. Limitations and Future Recommendations

Almost all research studies included in this review focused on women that were pregnant and started supplementation many weeks after conception. Studies that involve pre-conception use of supplements may provide even stronger evidence for benefits. We strongly recommend that future studies incorporate pre-conception use of prenatal supplements because many women start prenatal supplements several months or more prior to conception.

Although the present review focuses on mineral supplementation for women in the U.S., many of the research studies were conducted outside the U.S. Therefore, it was essential to consider the NHANE’s daily intake in the recommendations to better understand the vitamin and mineral status of U.S. women, as nutritional deficiencies in the U.S. may be different than in other parts of the world.

This paper involves an extensive review of many of the most relevant studies (including reviews and meta-analyses), but due to the scope of the field, it is not a systematic review, and it was not possible to review every published study for all minerals in a single paper. However, it is hoped that this review provides a broad overview of the field, and the evidenced-based recommendations made here can be reviewed in more detail for each mineral in future papers. It is clear that more research is needed, especially for minerals that have been researched less. These recommendations are based on the current evidence but should be expected to change somewhat as more research is conducted. The evidence for some of the recommendations is limited, but we think that there is enough evidence to make reasonable recommendations for all minerals. In general, at the level of supplementation recommended here, the potential benefits appear to far outweigh the rare potential risks. Hence, we believe it is ethically and scientifically justified to make these recommendations now rather than wait a decade or more for additional research.

This review has focused on nutritional supplementation, but improving diets is also an important goal. Prenatal supplements should be used to supplement but not replace a healthy diet.

## 5. Conclusions

Many aspects of maternal and infant health can be improved by prenatal supplementation at appropriate levels. This paper proposes a comprehensive set of evidence-based recommendations for the optimal level of each mineral for a prenatal supplement. However, mineral content of prenatal supplements varies widely, and a review of 188 prenatal supplements found that they often included only a subset of minerals, and often at low levels, so that only a small fraction of supplements met our recommendations. Thus, we hypothesize that following the proposed recommendations for mineral supplementation will reduce the rate of many maternal and infant health complications in the U.S.

## Figures and Tables

**Figure 1 nutrients-13-01849-f001:**
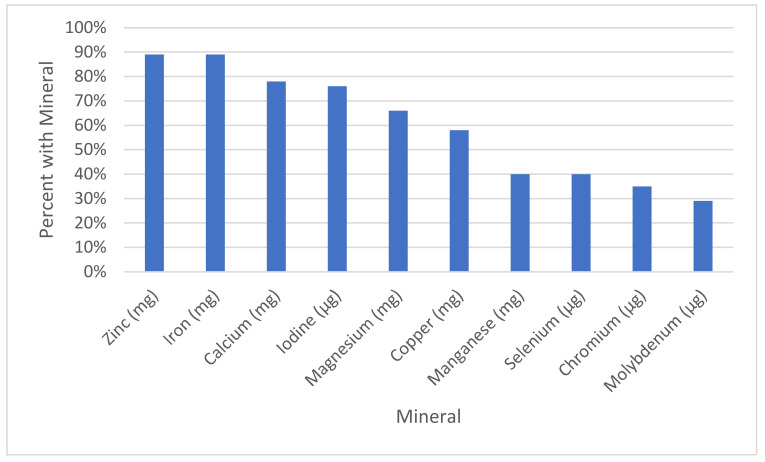
Percent of Supplements with this Mineral.

**Figure 2 nutrients-13-01849-f002:**
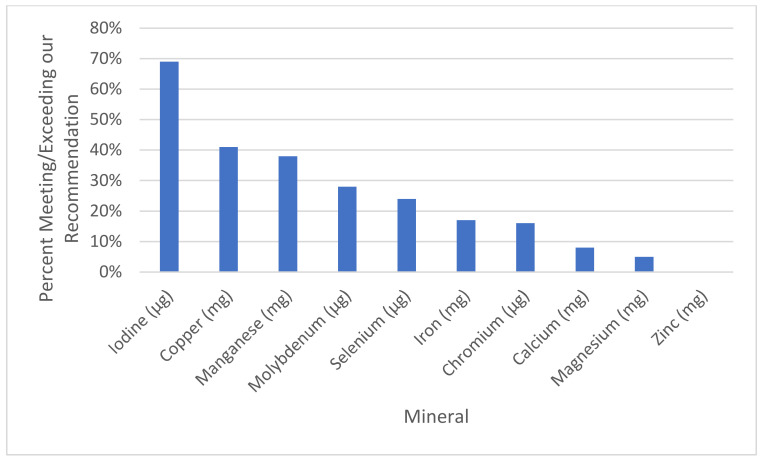
Percent of supplements meeting or exceeding our recommendation.

**Table 1 nutrients-13-01849-t001:** Relationship of maternal health complications to mineral status. Superscripts of M indicate meta-analyses, and R indicates a Review of multiple studies [3,7,8,10,11,14,15,16,17,20,24,26,27,28,30,31,32,33,35,36,37,39,40,49,94,95,96,97,100,102,103,104,105,116,124,126,127,128,130,132,141].

Maternal Outcome	Significant Evidence	Limited Evidence
Anemia	Iron [77,78,83,84,85,86,87] and [89] M	
Anxiety		Copper [39]
Blighted Ovum		Copper [31,33]
Depression		Copper [39]
Edema		Magnesium [102] Selenium [130]
Fertility (for women with Polycystic Ovary Syndrome)		Chromium [27,28]
Gestational Diabetes	Chromium [20,24,26]	Selenium [127] M, [128] R
Gestational Hypertension	Calcium [7] M, [8] R, [14,15,16] Magnesium [7] M, [100,103] Selenium [124,130] Zinc [7] M, [141] M	
Hypothyroidism	Selenium [124,130,132] Iodine [49]	Selenium [132]
Infection		Selenium [126]
Leg Cramps		Magnesium [97]
Maternal Death/Serious Morbidity		Calcium [8] R
Maternal Hospitalization	Magnesium [94,104,105]	
Miscarriage	Copper [31,32,33] Selenium [128] R	
Postpartum Depression		Chromium [30] Selenium [130]
Preeclampsia	Calcium [3], [8] R, [10,11,14,17] Copper [36] M, [37] M Magnesium [95,96,97] Selenium [124,128,130] Zinc [7] M, [141] M	Iodine [40] Manganese (high levels are a problem) [116]
Premature Rupture of Membranes (PROM)		Copper [35]

**Table 2 nutrients-13-01849-t002:** Relationship of infant health problems to maternal mineral status [1,2,8,12,13,33,34,41,42,43,49,59,66,80,81,83,84,85,86,87,88,97,98,100,104,105,106,107,109,111,112,113,114,117,118,120,128,135,138,142,145,147].

Infant Outcome	Significant Evidence	Limited Evidence
ADD		Iodine [59]
ADHD	Iodine [66] R	
Anemia	Iron [83,84,85,86,87,88]	
Anencephaly		Copper [33] Iron [82]
Apgar Score	Magnesium [100] R	Zinc [138]
Asthma		Zinc [143]
Autism	Calcium [12,13] Iodine [41,42]	Iron [80] Molybdenum [120]
Birth Weight	Magnesium [104,105,106] Manganese [111,112,114]	Calcium [8] R Selenium [128] R Zinc [135,138]
Central Nervous System (CNS) Malformations		Copper [34]
Cerebral Palsy	Magnesium [109] R	
Congenital Diaphragmatic Hernia		Selenium [128]
Dental Cavities		Calcium [8] R
Fetal Distress		Zinc [144]
Fetal Growth Restriction		Magnesium [111]
High Blood Pressure		Calcium [8] R
Hypothyroidism	Iodine [43] R, [49]	
Impetigo		Zinc [150]
Infant Mortality		Iodine [54]
Intellectual Disability	Iodine [43,54,55,56,69] Manganese [117,118]	
Intrauterine Growth Restriction		Selenium [128] R Zinc [138]
Long Delivery		Selenium [145]
Neonatal Intensive Care Unit Admissions		Calcium [8] R
Neural Tube Defects	Selenium [128] R	
Orofacial Cleft		Iron [81] Magnesium [81]
Preterm Birth	Calcium [8] R Magnesium [97,98,104,105,106,107] Zinc [147] M, [138]	Manganese [113]
Rickets	Calcium [1,2]	
Wheeze	Zinc [142] M	

Table 2 demonstrates the significance of a mineral in conditions that can affect an infant. Substantial nutritional interference is regarding minerals that have two or more research articles that support the correlation, while possible nutritional interference only has one supporting research article.

**Table 3 nutrients-13-01849-t003:** Relationship of minerals to maternal health problems/benefits [7,8,10,11,12,13,14,15,16,17,20,26,27,30,31,32,33,35,36,37,39,40,43,49,77,78,83,84,85,87,89,94,95,96,97,100,103,104,105,116,124,127,128,130,132,141].

Mineral	Significant Evidence	Limited Evidence
Calcium	Gestational Hypertension [7] M, [8] R, [14,15,16] Maternal Death [8] R Preeclampsia [8] R, [10,11,14,17] Pregnancy Induced Hypertension [7] M, [8] R, [10,11,14,15,16]	
Chromium	Gestational Diabetes [20,26]	Postpartum Depression [30] Fertility in Women with Polycystic Ovary Syndrome [27]
Copper	Miscarriages [31,32,33] Preeclampsia [36] M, [37] M	Anxiety [39] Depression [39] PROM [35]
Iodine	Hypothyroidism [43,49]	Preeclampsia [40]
Iron	Anemia [77,78,83,84,85,87,89]	
Magnesium	Gestational Hypertension [7] M, [42,100,103] Maternal Hospitalization [94,104,105] Preeclampsia [95,96,97]	Leg Cramps [97]
Manganese		Preeclampsia (high levels are a problem) [116]
Molybdenum		
Selenium	Miscarriage [128] R Preeclampsia [124,128,130] Gestational Hypertension [124,130]	Gestational Diabetes [127] M, [128] R Hypothyroidism [124,130,132] Postpartum Depression [130] Postpartum Thyroid Deficiency [132]
Zinc	Preeclampsia [7] M, [141] M Gestational Hypertension [7] M, [141] M	

**Table 4 nutrients-13-01849-t004:** Relationship of maternal mineral status to infant health problems/benefits [1,2,8,12,13,33,34,43,44,54,55,56,66,80,81,82,85,86,87,88,89,97,98,100,104,105,106,107,111,112,113,114,117,118,126,128,138,142,143,147,150].

Mineral	Significant Evidence	Limited Evidence
Calcium	Autism [12,13] Preterm Birth [8] R Rickets [1,2]	High Blood Pressure [8] R Dental Cavities [8] R Low Birthweight [8] R
Chromium		
Copper		Anencephaly [33] CNS Malformations [34]
Iodine	ADHD [66] R Hypothyroidism [43] R, [44] Intellectual Disability [43] R, [54,55,56]	Mortality [54] IQ [63] M Stillbirth [54]
Iron	Anemia [85,86,87,88,89]	Anencephaly [82] Autism [80] Orofacial Cleft [81]
Magnesium	Apgar Score [100] R Birth Weight [104,105,106] Cerebral Palsy [109] R Preterm Birth [97,98,104,105,106,107]	Orofacial Cleft [81]
Manganese	Birth Weight [111,112,114] Intellectual Disability [117,118]	Preterm Birth [113]
Selenium	Neural Tube Defects [138] R	Birthweight [128] R Congenital Diaphragmatic Hernia [128] R Intellectual Disability [126] Intrauterine Growth Restriction [128] R
Zinc	Preterm Birth [138], [147] M Wheeze [142] M	Apgar Scores [138] Birth Weight [138] Duration of Gestation [138] Impetigo [150] Asthma [143]

**Table 5 nutrients-13-01849-t005:** Recommendations for content of prenatal supplements and other relevant data for comparison.

Nutrient	Our Recommendation	RDA Recommendation for Total Daily Intake for Pregnant Women	Daily Intake (Women Aged 20–39) per NHANES Unless Otherwise Noted	Tolerable Upper Limit for Pregnant Women	Change during Pregnancy
Calcium	550 mg (1000 mg for those with greater risk of preeclampsia)	1000 mg	872 mg	1000 mg	Decreases
Chromium	100 mg (200 mg for women with diabetes)	30 μg	23 to 29 μg [19]	-	Possibly decreases
Copper	1.3 mg	1 mg	1.1 mg	10 mg	Levels increase, but low levels associated with health complications
Iodine	150 μg/day	220 μg	160 μg *	1100 μg	Possibly decreases
Iron	30 mg 1st trimester, 60 mg 2nd trimester and 3rd trimester; up to 60 mg 3× day in extreme cases	27 mg	12.2 mg	45 mg	Decreases
Magnesium	350 mg	400 mg	269 mg	400 mg	Decreases
Manganese	1 mg	2.0 mg	2.3 mg [122]	11 mg	Increases, but low levels are associated with health complications
Molybdenum	25 μg	50 μg	76 μg [122]	2000 μg	Unknown
Selenium	60 μg	70 μg	97 μg	400 mg	Decreases
Zinc	30 mg	11 mg	9.4 mg	40 mg	Decreases

* Intake based on NHANES study which evaluated women aged 15–44 years. RDA: Recommended Dietary Allowance.

**Table 6 nutrients-13-01849-t006:** Comparison of prenatal supplements on market vs. our recommendations.

Mineral	Our Reccomendation	% of Sups with This Mineral (Out of 188)	% Meeting or Exceeding Rec.	Average All	Average of Those with the Mineral	% of the Average of All Supplements Divided by Our Recommendation	% of the Average of Those with Supplements Divided by Our Recommendation	Range
Calcium (mg)	550	78% (146)	8% (15)	204.2 ± 233.8	262.9 ± 233.6	37%	48%	0–1300
Chromium (μg)	100	35% (66)	16% (31)	28.9 ± 51.6	82.3 ± 56.2	29%	82%	0–200
Copper (mg)	1.3	58% (109)	41% (77)	0.9 ± 0.9	1.5 ± 0.6	67%	115%	0–2
Iodine (μg)	150	76% (143)	69% (129)	138.9 ± 98.0	182.6 ± 67.6	93%	122%	0–316
Iron (mg)	30	89% (168)	17% (32)	24.2 ± 15.9	27.1 ± 14.2	81%	90%	0–91.5
Magnesium (mg)	350	66% (124)	5% (9)	75.0 ± 110.1	113.7 ± 117.8	21%	32%	0–500
Manganese (mg)	1	40% (76)	38% (71)	1.1 ± 1.7	3.3 ± 1.6	100%	273%	0–40
Molybdenum (μg)	25	29% (55)	28% (53)	16.1 ± 28.0	55.1 ± 22.8	64%	220%	0–100
Selenium (μg)	60	40% (76)	24% (46)	27.3 ± 43.6	67.4 ± 44.3	45%	112%	0–200
Zinc (mg)	30	89% (167)	0% (0)	13.3 ± 7.7	15.0 ± 6.5	44%	50%	0–25

## Data Availability

Information on prenatal supplements currently on the market is available upon request.

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
