# Peer review of "Evidence-Based Recommendations for an Optimal Prenatal Supplement for Women in the U.S., Part Two: Minerals"

_nutrients, 2021, doi:10.3390/nu13061849_

Round 1
Reviewer 1 Report
The article in view of publication written by Adams et al, is a matter of real importance in terms of public health. The health of the pregnant woman is a fundamental stage for the health of the fetus, the newborn and the future adult. It is known that events that occur during the fetal stage can have an impact on adult life (fetal programming)
I congratulate the authors for the work that is both exhaustive and synthetic.
However, I have some points that I would like the authors to clarify. Specifically in the chapters dedicated to calcium and iodine
Major points
Page 4, lines 172-173. If the authors indicate that high doses of calcium can cause HELLP syndrome, they could be more specific and give a very specific dose threshold.
Page 4, lines 184-185. In agreement with my previous comment. The authors should correct the phrase "at least 1000 mg calcium": what is the maximum amount that does not expose to a possible HELLP syndrome
Pages 7, lines 305-398, Chapter 3.4. The authors emphasize the beneficial effects of iodine during pregnancy, but they neglect the possible harmful effects of excessive iodine supplementation during pregnancy. It's known that both inadequate and excessive iodine intakes are associated with increases in serum thyroglobulin values. An excess of iodide inhibits the synthesis of thyroid hormone. This is known as the acute Wolff-Chaikoff effect. The proposed mechanism is the formation of iodopeptides which temporarily inhibit thyroid peroxidase (TPO) mRNA and protein synthesis and therefore thyroglobulin iodination.
Furthermore, an excessive iodine intake (oral or transdermal) during pregnancy increases the risk of hypothyroidism, hyperthyroidism or goiter in newborns (Bartalenaet al. 2001; Caronet al. 2006; Connelly et al. 2012 ; Glinoer 1997; Nishiyamaet al. 2004; Pennington 1990; Sanget al. 2012; Trumpffet al. 2013; Rodriguez-Diaz 2020).
What would be the maximum safe dose?
Minor points
Page 11, lines 532-535. Please, harmonize the characters with the rest of the document
Page 14, lines 649, 670, 671. Please, harmonize the characters with the rest of the document
Page 17, line 742. Change the title of the table, put the word "benefit" in evidence
Page 40, lines 764-773. Please, harmonize the characters with the rest of the document
Author Response
The article in view of publication written by Adams et al, is a matter of real importance in terms of public health. The health of the pregnant woman is a fundamental stage for the health of the fetus, the newborn and the future adult. It is known that events that occur during the fetal stage can have an impact on adult life (fetal programming)
I congratulate the authors for the work that is both exhaustive and synthetic.
Response: We thank the reviewer for their positive comments.
However, I have some points that I would like the authors to clarify. Specifically in the chapters dedicated to calcium and iodine
Major points
Page 4, lines 172-173. If the authors indicate that high doses of calcium can cause HELLP syndrome, they could be more specific and give a very specific dose threshold.
Response: We have added the two dosages (1500 ang 2000 mg of elemental calcium) in the two studies which reported a risk of HELLP syndrome. See line 142 - 143
Page 4, lines 184-185. In agreement with my previous comment. The authors should correct the phrase "at least 1000 mg calcium": what is the maximum amount that does not expose to a possible HELLP syndrome
Response: We have clarified the dosages (1500-2000 mg/day) – see line 181
Pages 7, lines 305-398, Chapter 3.4. The authors emphasize the beneficial effects of iodine during pregnancy, but they neglect the possible harmful effects of excessive iodine supplementation during pregnancy. It's known that both inadequate and excessive iodine intakes are associated with increases in serum thyroglobulin values. An excess of iodide inhibits the synthesis of thyroid hormone. This is known as the acute Wolff-Chaikoff effect. The proposed mechanism is the formation of iodopeptides which temporarily inhibit thyroid peroxidase (TPO) mRNA and protein synthesis and therefore thyroglobulin iodination.
Response: Thank you for the helpful suggestion. We have added a discussion of this – see lines 397 – 403.
Furthermore, an excessive iodine intake (oral or transdermal) during pregnancy increases the risk of hypothyroidism, hyperthyroidism or goiter in newborns (Bartalenaet al. 2001; Caronet al. 2006; Connelly et al. 2012 ; Glinoer 1997; Nishiyamaet al. 2004; Pennington 1990; Sanget al. 2012; Trumpffet al. 2013; Rodriguez-Diaz 2020).
Response: We have added a discussion of this – see lines 304 - 410
Also, due to this change, and new data we found on NHANES intake of iodine, we have altered our recommendations for supplemental intake of iodine. See changes in tables 5 and 6, and in the text. – see lines 415 – 419
What would be the maximum safe dose?
Response: We added the statement “ The Tolerable Upper Intake is set at 1100 mcg/day for all people including pregnant women, the WHO has recommended that total daily intake above 500 mcg/day may be excessive.” See lines 411 - 413
Minor points
Page 11, lines 532-535. Please, harmonize the characters with the rest of the document
Response: We have harmonized the characters as requested and added more detail.
14, lines 649, 670, 671. Please, harmonize the characters with the rest of the document
Response: We have harmonized the characters and added relative rates to lines 700 to 708, and confidence interval to line 728.
Page 17, line 742. Change the title of the table, put the word "benefit" in evidence
Response: We added the word “benefits” to the table caption for tables 3 and 4
Page 40, lines 764-773. Please, harmonize the characters with the rest of the document
Response: We have corrected the statement about the decrease in minerals during pregnancy
Reviewer 2 Report
Overall, it is an important and well-written paper. However, it is difficult to see what kind of studies are included in the review. Tables 1 - 4 are marked with R or M, respectively, to show if it is a review or meta-analysis but for the other studies mentioned, we do not know what study design they have.
In table 4 (selenium), this author, Mariath 2011 M, is underlined. Why?
The authors are given this statement in the Methods section: "Since the research literature is vast, a systematic review of all studies would require a separate paper on each mineral; instead, we focus on the most relevant articles that we found from keyword searches of PubMed, and forward and backward citation searches of the most relevant articles". What do you consider as relevant articles? You have included articles from as far back as 1976. Wouldn't it be more relevant to include the newest articles? The most recent article included is from 2019.
Also, there is no information about how many participants in each study. I would like to see an overview table with each study included in this review. In general, it is difficult to follow how the authors have selected the articles included in their review: what were the inclusion or exclusion criteria? Based on this missing information it is difficult to judge the quality of this review.
Author Response
Overall, it is an important and well-written paper. However, it is difficult to see what kind of studies are included in the review. Tables 1 - 4 are marked with R or M, respectively, to show if it is a review or meta-analysis but for the other studies mentioned, we do not know what study design they have.
Response: We thank the reviewer for their positive comments. For Tables 1-4 we have clarified the nature of the other studies.
In table 4 (selenium), this author, Mariath 2011 M, is underlined. Why?
Response: This was a typo, and has been corrected. Thank you.
The authors are given this statement in the Methods section: "Since the research literature is vast, a systematic review of all studies would require a separate paper on each mineral; instead, we focus on the most relevant articles that we found from keyword searches of PubMed, and forward and backward citation searches of the most relevant articles". What do you consider as relevant articles? You have included articles from as far back as 1976. Wouldn't it be more relevant to include the newest articles? The most recent article included is from 2019.
Response: Thanks for asking for clarification. The primary focus of this review is on articles that provided insight into optimal dosage, such as treatment studies on the effect of different doses on outcomes and biomarkers. This has been added to the text – please see lines 80-82.
Re. the year of the studies, we did include some older studies, but we have included studies up to 2019/2020. Most of the review was done in 2019, and then we did most of the analysis and writing in 2020. However, we have just done an updated search and added several newer references.
Also, there is no information about how many participants in each study. I would like to see an overview table with each study included in this review. In general, it is difficult to follow how the authors have selected the articles included in their review: what were the inclusion or exclusion criteria? Based on this missing information it is difficult to judge the quality of this review.
Response: Thanks for the suggestion: We have added an overview table of all the articles included in this review, including the number of participants, the country the study was conducted in, and the type of study. An unusual feature and strength of this study is that we do not rely on a single type of study, such as randomized clinical trials, but also consider observational studies (such as correlations of biomarkers with outcomes and effect of supplementation on biomarkers), and we have added that information to the new table as suggested.
Round 2
Reviewer 2 Report
Thank you for this nice review. I have no further objections to your study.